# Effectiveness of intravenous lidocaine in preventing postoperative nausea and vomiting in pediatric patients: A systematic review and meta-analysis

**Daisuke Nakajima**[1]*, **Hiromasa Kawakami**[2], **Takahiro Mihara**[3,4], **Hitoshi Sato**[2], **Takahisa Goto**[4]

**1** Intensive Care Department, Yokohama City University Medical Center, Yokohama, Japan, **2** Department of Anesthesiology, Yokohama City University Medical Center, Yokohama, Japan, **3** Education and Training Department, YCU Center for Novel and Exploratory Clinical Trials, Yokohama City University Hospital, Yokohama, Japan, **4** Department of Anesthesiology, School of Medicine, Yokohama City University, Yokohama, Japan

* damikami3658@gmail.com

**Data Availability Statement:** All relevant data are within the manuscript and its Supporting Information files.

## Abstract

### Background

Intravenous lidocaine in adults undergoing general anesthesia has been shown to reduce the incidence of postoperative nausea and vomiting (PONV). However, the anti-postoperative vomiting (POV) effect of lidocaine in pediatric patients remains unclear. We conducted a systematic review and meta-analysis with Trial Sequential Analysis to evaluate the effect of intravenous lidocaine on prevention of POV/PONV.

### Methods

Six databases including trial registration sites were searched. Randomized clinical trials evaluating the incidence of POV/PONV after intravenous lidocaine compared with control were included. The primary outcome was the incidence of POV within 24 hours after general anesthesia. The incidence of POV was combined as a risk ratio with 95% confidence interval using a random-effect model. We used the $I^2$ to assess heterogeneity. We evaluated the quality of trials using the Cochrane methodology, and we assessed quality of evidence using the Grading of Recommendation Assessment, Development, and Evaluation approach. We also assessed adverse events.

### Results and discussion

Six trials with 849 patients were included, of whom 433 received intravenous lidocaine. Three trials evaluated the incidence of POV, and 3 evaluated the incidence of PONV. The overall incidence of POV within 24 hours after anesthesia was 45.9% in the lidocaine group and 63.4% in the control group (risk ratio, 0.73; 95% confidence interval, 0.53–1.00; $I^2$ = 32%; p = 0.05). The incidence of PONV within 24 hours after anesthesia was 3.73% in the lidocaine group and 4.87% in the control group (RR, 0.76; 95% CI, 0.36–1.59; $I^2$ = 0%; p =

**Funding:** We received only the departmental funding from Yokohama City University Graduate School of Medicine. The funding source had no role in the study design, data collection, and analysis, decision to publish, or preparation of the manuscript.

**Competing interests:** The authors have declared that no competing interests exist.

0.47). The quality of evidence was downgraded to "very low" due to the study designs, inconsistency, imprecision, and possible publication bias.

## Conclusion

Our meta-analysis suggests that intravenous lidocaine infusion may reduce the incidence of POV, however, the evidence quality was "very low." Further trials with a low risk of bias are necessary.

## Introduction

Postoperative vomiting (POV) is a particularly important complication in pediatric patients undergoing general anesthesia. Its reported frequency in pediatric patients ranges from 13% to 42%, which is approximately twice as frequent as in adults, and the frequency increases to 30% to 80% in high POV risk pediatric patients [1,2]. POV is the leading cause of parental dissatisfaction after pediatric surgery and has always been considered as one of the scabrous problems after pediatric general anesthesia. [3]. Severe POV may result in extended hospital stays, unexpected admissions after day-case surgery, and high medical costs [3–5]. Therefore, a number of pharmacological treatments such as ondansetron and dexamethasone have been studied to prevent POV in pediatric patients [6,7]. However, these pharmacological treatments are financially costly or have several adverse effects such as QT interval prolongation and postoperative bleeding [8].

Lidocaine is a common adjuvant for pediatric general anesthesia [9], and some studies have demonstrated that it prevents perioperative adverse events such as opioid-induced cough, laryngospasm, and propofol-induced pain in pediatric surgical patients [10–13]. There is evidence suggesting that the use of intravenous lidocaine in adult patients undergoing general anesthesia could reduce the incidence of postoperative nausea and vomiting (PONV) [14–16]. The anti-PONV mechanism of lidocaine is unclear, but it might be due to a gastrointestinal recovery or an opioid-sparing effect [16,17]. However, the anti-POV/PONV effect of lidocaine in pediatric patients remains unclear. The purpose of this meta-analysis was to assess the anti-POV/PONV effect and possible adverse events of intravenous lidocaine in pediatric surgical patients.

## Methods

We conducted a systematic review with meta-analysis and Trial Sequential Analysis (TSA). This meta-analysis was performed according to the recommendations of the Preferred Reporting Items for Systematic Reviews and Meta-Analyses (PRISMA) statement [18] and the Cochrane Handbook [19]. Our study protocol and methods were pre-specified and are registered on PROSPERO (CRD42018099029).

### Search strategy

We searched Pubmed, EMBASE, Cochrane Central Register of Controlled Trials, and Web of Science databases. We also searched clinicaltrials.gov, and University Hospital Medical Information Clinical Trials Registry. The last search was on 1 May 2019. We also searched related reviews and reference lists. The PubMed search strategy is provided in the S1 Text.

Two authors (D.N. and H.K.) independently assessed the suitability of titles and abstracts of the studies identified by the search strategies to exclude irrelevant articles. We retrieved the full-text versions of potentially relevant studies selected by at least one author, and those that met the inclusion criteria were then examined separately. The discrepancies were resolved by consensus through discussion between the two authors. We searched for randomized clinical trials (RCTs) that evaluated the incidence of POV/PONV after the intravenous lidocaine compared with a placebo or no medication in pediatric patients undergoing general anesthesia. We excluded studies that did not evaluate the incidence of POV/PONV, in which the subjects were not pediatric patients (aged more than 18), and in which lidocaine was not administered intravenously. We also excluded data from case reports, observational studies, comments, letters to the editor, reviews, and animal studies. Eligibility was not restricted by language or type of surgery.

## Outcomes

The primary outcome was the incidence of incidence of POV within 24 hours after anesthesia. Secondary outcomes included overall incidence of PONV, early (0–6 hours) and late (6–24 hours) POV/PONV, serum lidocaine concentration before and after surgery, the need for anti-emetic rescue medication, severity of POV/PONV if measured with a numeric rating scale or visual analogue scale, and adverse events of lidocaine such as seizure, arrhythmias, or allergic reactions.

## Data collection

A data-collection sheet was created, which included: (i) the number of patients in the study; (ii) age; (iii) American Society of Anesthesiologists (ASA)-Physical Status; (iv) risk factors for PONV (history of PONV or motion sickness); (v) type of surgery; (vi) dose of lidocaine; (vii) timing of administration of lidocaine; (viii) number of cases of POV in the early, late, and next day period; (ix) number of cases of PONV in the early, late, and next day period; (x) need for rescue anti-emetics in the early, late, and next day period; (xi) severity of POV/PONV; (xii) adverse events such as seizures, arrhythmia, or allergic reactions. Two authors (D.N. and H.K.) independently extracted the data from the included studies using a piloted form and cross-checked the data.

## Assessment of risk of bias

We followed the Cochrane Handbook for Systematic Reviews of Interventions and evaluated the trial risk of bias [19]. We evaluated the risk of bias in the following seven potential sources: "sequence generation," "allocation sequence concealment," "the blinding of patients or health care providers," "the blinding of outcome assessors," "incomplete outcome data," "selective outcome reporting," and "other bias." The risk of bias was classified as following: "low," "high," or "unclear." The data were subsequently cross-checked by two authors (D.N. and H. K.). When there was a discrepancy in the evaluations of bias, the two authors discussed their evaluations and reached a consensus. We considered a trial as having a high risk of bias if one or more risks of bias was classified unclear or high.

## Statistical analysis

We analyzed the data using Review Manager, version 5.3.5 (RevMan, The Cochrane Collaboration, Oxford, United Kingdom). We compared the incidence of POV/PONV with the risk ratio (RR). We summarized the RR with a 95% confidence interval (CI). If the 95% CI included

1, we considered the difference not to be statistically significant. We used a random-effect model (DerSimonian and Laird methods[20]) to combine the results of trials. Heterogeneity was quantified with the $I^2$ statistic. Forest plot was used to visualize and evaluate the results of trials. Small study effect was assessed using a funnel plot if the number of trials was greater than nine. Sensitivity analysis was performed for the primary outcome based on trials with a low risk of bias. We also performed Trial Sequential Analysis (TSA) for the primary outcome to reduce false-positive results caused by multiple testing and sparse data. We calculated quantified TSA monitoring boundaries (i.e. Monitoring boundaries for meta-analysis) and required information size (RIS), and adjusted CIs. Risk of type 1 error was maintained at 5% with a power of 90%. We considered a reduction of RR by 25% to be clinically meaningful. If the TSA-adjusted CI included a value of 1, the difference was considered not statistically significant. We conducted the TSA using TSA viewer version 0.9.5.10 Beta (Copenhagen Trial Unit, Copenhagen, Denmark; www.ctu.dk/tsa).

### Assessment of quality of evidence

We used the Grading of Recommendations Assessment, Development, and Evaluation (GRADE) approach [21,22] to evaluate the quality of evidence. We evaluated following domains: risk of bias, inconsistency, indirectness, imprecision of the results, and publication bias. The quality of evidence for the main outcome was classified as very low, low, moderate, or high. We used GRADEpro GDT (https://gradepro.org/) to create a summary of finding table.

## Results

### Search selection and study characteristics

In the initial search of the databases, 2099 articles were identified. We examined the full texts of 75 articles in detail. We contacted the journal offices, but we could not obtain the full text of one article [23]. Of those, 6 trials with 849 patients were included, and 433 of them received intravenous lidocaine (Fig 1). Five of the included articles were written in English [13,24–27], and one in Turkish [28]. Three trials evaluated the incidence of POV [24–26], and 3 trials evaluated the incidence of PONV [13,27,28]. The characteristics of the randomized clinical trials included in this systematic review are shown in Table 1. All trials compared intravenous lidocaine with a placebo. The bolus dose of lidocaine ranged from 1.5 to 2 mg.kg$^{-1}$. One trial used 1.5 mg.kg$^{-1}$ bolus dose and 2 mg.kg$^{-1}$·h$^{-1}$ continuous lidocaine infusion for the mean duration of 41.7 minutes [24]. The patient age ranged from 2 months to 14 years.

### Intervention effects

Three trials evaluated POV as primary outcome [24–26], and the overall incidence of POV within 24 hours after anesthesia was 45.9% in the lidocaine group and 63.4% in the control group (RR, 0.73; 95% CI, 0.53–1.00; $I^2$ = 32%; p = 0.05). The combined results are shown in Fig 2. One trial evaluated the incidence of PONV as primary outcome [28], and 2 trials evaluated the incidence of PONV as one of adverse events [13,27]. The incidence of PONV within 24 hours after anesthesia was 3.73% in the lidocaine group and 4.87% in the control group (RR, 0.76; 95% CI, 0.36–1.59; $I^2$ = 0%; p = 0.47). The combined results are shown in Fig 3. Only 1 trial evaluated the occurrence of POV/PONV by time phases (early, late, and next day) [25].

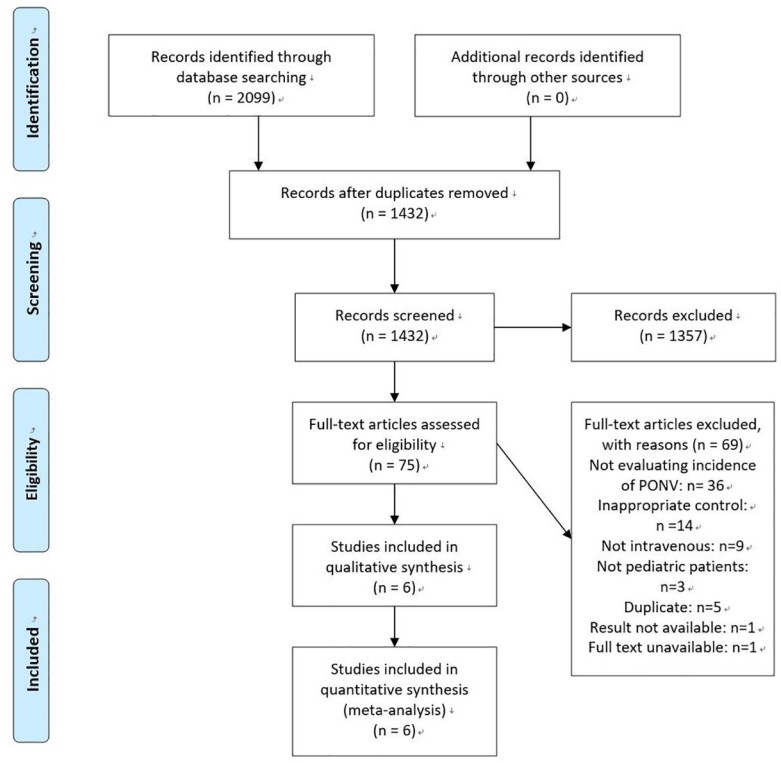

**Fig 1. PRISMA flow diagram.**

## The risk of bias of the included trials

The risk of bias in the included trials is summarized in Fig 4. We considered 2 trials to be at low risk of bias, while 4 were at high risk of bias.

## Small-study effects

We could not conduct an asymmetry test for the funnel plot because only 6 trials were included.

**Table 1. Characteristics of included trials.**

| Source | ASA-PS | age (protocol) | Surgery | Anesthesia agent used | Bolus dose of study drug | Continuous dose of study drug | Timing of Study Drug (bolus) |
|---|---|---|---|---|---|---|---|
| Bilotta 2006 [13] | 1–2 | 2 months–10 yr | Bone biopsy | propofol | 2 mg.kg$^{-1}$ | none | 1 min before the induction of anesthesia |
| Echevarria 2018[24] | 1–2 | 2 yr–12 yr | elective tonsillectomy | sevoflurane, N$_2$O | 1.5 mg.kg$^{-1}$ | 2 mg.kg-1.h$^{-1}$ | induction |
| Tramer 1998 [25] | not defined | 3 yr–6 yr | strabismus surgery | propofol | 2 mg.kg$^{-1}$ | none | 10 min before start of surgery |
| Turkoglu 1995 [28] | 1–2 | 3 yr–14 yr | strabismus surgery | halothane | 1.5 mg.kg$^{-1}$ | none | before the induction of anesthesia |
| Warner 1988 [26] | 1 | 18 months–7 yr | strabismus surgery | halothane, N$_2$O | 2 mg.kg$^{-1}$ | none | 90 s prior to laryngoscopy |
| Young 2005 [27] | 1–2 | 2 yr–7 yr | lower abdominal and genital surgery | sevoflurane, N$_2$O | 1.5 mg.kg$^{-1}$ | none | 5 min before discontinuation of their anesthetic |

ASA-PS: American Society of Anesthesiologists Physical Status

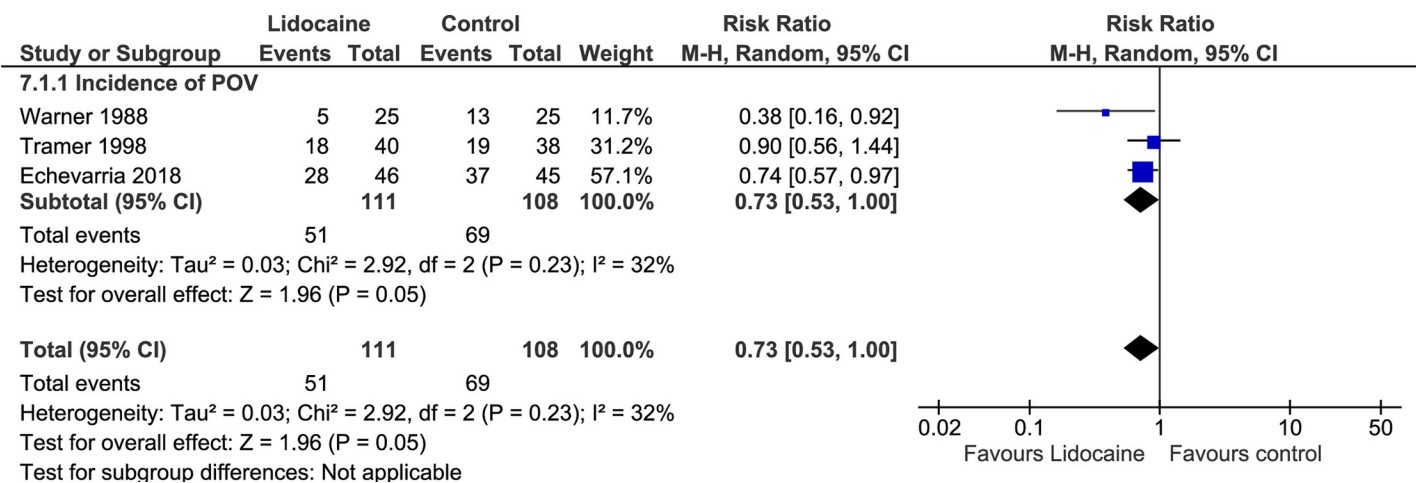

**Fig 2. Forest plot of the incidence of POV. POV, postoperative vomiting; CI, confidence interval.**

## Sensitivity analysis

We did not conduct a sensitivity analysis according to the risk of bias because only 1 trial with low risk of bias that evaluated POV as primary outcome was included.

## Trial sequential analysis

Trial sequential analysis for 'the incidence of POV within 24 hours' showed that the estimated required information size was 821; however, the accrued information size reached was only 219 (26.7%). The Z curve did not cross the TSA monitoring boundary or reach the required information size (Fig 5). This indicates that sufficient data have not been accumulated to determine conclusively whether intravenous lidocaine has anti-POV effect in pediatric surgical patients.

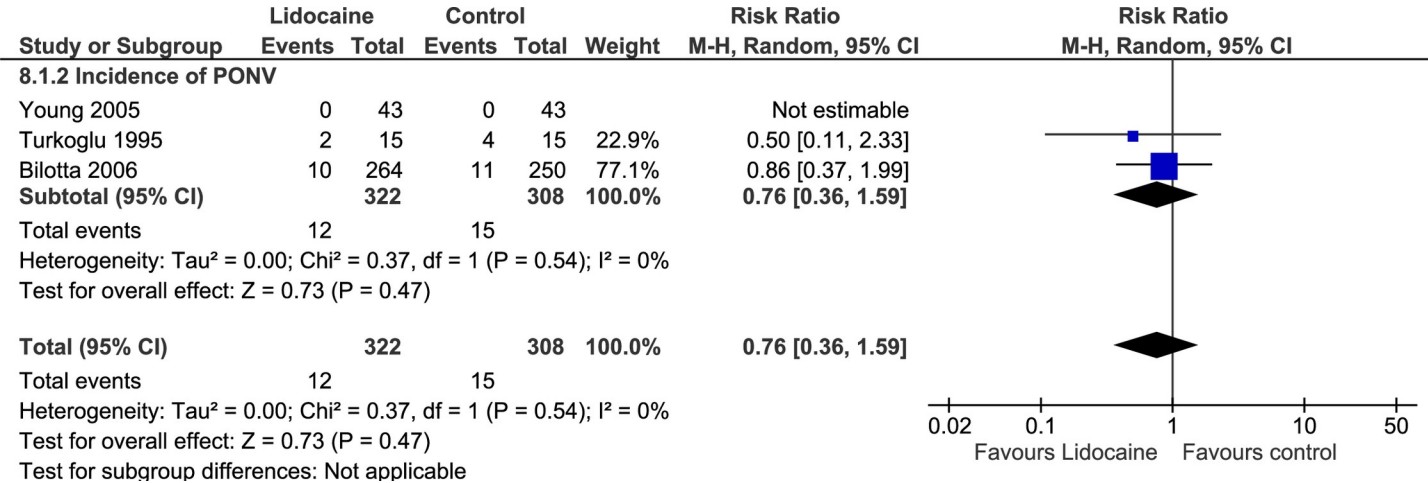

**Fig 3. Forest plot of the incidence of PONV. PONV, postoperative of nausea and vomiting; CI, confidence interval.**

**Fig 4. The risk of bias of the included trials.**

## Quality of the evidence

We evaluated the quality of evidence using the GRADE system. The evidence quality of 'the incidence of POV within 24 hours' was very low because there were limitations in study designs, imprecision, and possible publication bias. Inconsistency and indirectness were not detected (S2 Table).

## Adverse events

No adverse events such as seizures, arrhythmias, or allergic reactions were reported. Serum lidocaine concentration was reported in one trial [24], and the median lidocaine plasma concentration was 3.94 μg.ml$^{-1}$ (range: 0.87 to 4.88), which was below the toxicity threshold of 5 μg.ml$^{-1}$ [29].

## Other outcomes

Two trials reported the need for rescue antiemetic medication [24,25]. The incidence was lower in the intravenous lidocaine group (RR, 0.31; 95% CI, 0.10–0.92; $I^2$ = 0%). The combined results are shown in Fig 6. Severity of POV/PONV was not reported in any of the included trials. Opioid consumption was evaluated in 2 trials (one used fentanyl [24] and the other used alfentanil [25]), and it was not statistically different.

## Discussion

Our meta-analysis demonstrated that intravenous lidocaine may reduce the incidence of POV in pediatric patients undergoing general anesthesia. Intravenous lidocaine may also reduce the incidence of PONV and the need for antiemetic rescue medication. However, we should

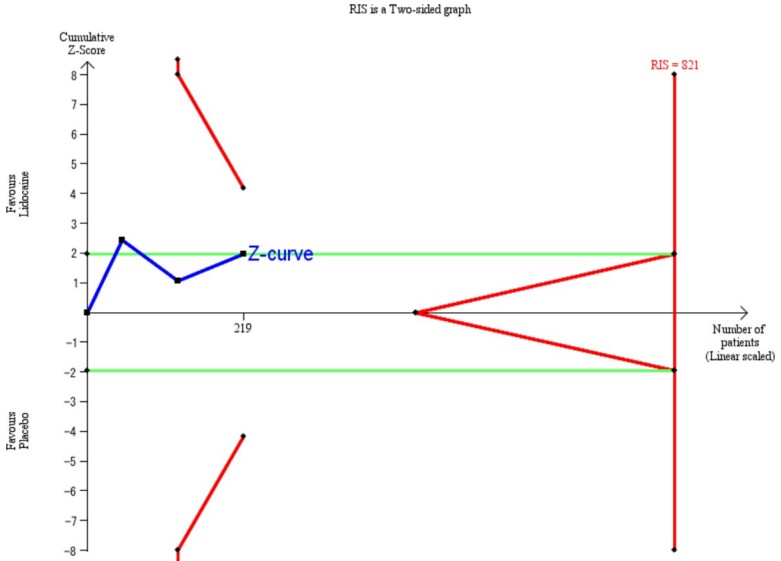

**Fig 5. Trial sequential analysis of effect of intravenous lidocaine on prevention of POV.** The risk of type 1 errors was set at 0.05 with a power of 0.9 when the Trial Sequential Analysis was performed. The variance was calculated from the data obtained from the included trials. A clinically significant reduction in risk ratio was set at 0.25.

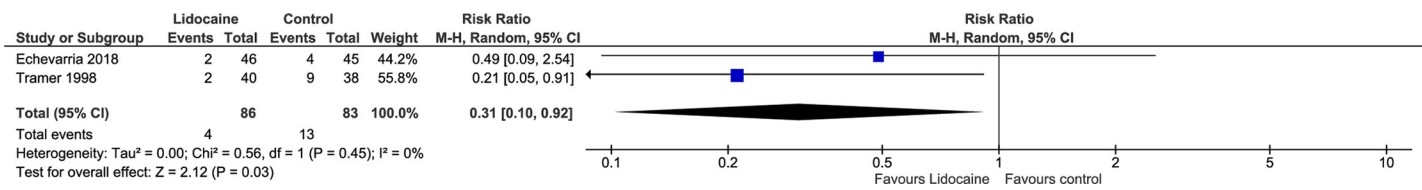

**Fig 6. Forest plot of the incidence of need for antiemetic rescue medication.** CI, confidence interval.

consider this analysis as hypothesis generative because the quality of the assessed evidence based on the GRADE approach was "very low".

We downgraded the GRADE due to study designs, imprecision, and possible publication bias, and evaluated the GRADE as "very low". According to TSA, the number of included patients in our meta-analysis reached only 26.7% of RIS. The TSA also revealed that the Z-curve did not cross the trial sequential monitoring boundary for benefit and the TSA-adjusted CI was wide. Considerable heterogeneity does not exist among our trials.

In our included trials, the incidence of POV/PONV in the control group in 4 trials that evaluated POV/PONV as a primary outcome [24–26,28] was excessively different from that in 2 trials that evaluated PONV as one of the adverse events. Two reasons can partly explain this. First, the incidence of PONV is greatly affected by the type of surgery, particularly in pediatric patients [30–32]; the patients in the 2 trials in which the incidence of PONV was low underwent surgery with low risk of PONV, whereas those in the other 4 trials underwent surgery with high risk of PONV such as strabismus surgery and tonsillectomy [6,30]. Second, the occurrence of PONV might not have been adequately evaluated in the 2 trials because it was evaluated as one of the adverse events.

Possible mechanisms of the anti-PONV effect of intravenous lidocaine in adult patients are a gastrointestinal recovery and an opioid-sparing effect [16,17]. In our included trials, 2 evaluated opioid consumption during the intraoperative or postoperative period [24,25], and found that it was not statistically different. The absence of significant differences might be due to the small sample size, but we could not conduct a meta-analysis concerning opioid consumption due to the different types of opioids used. Other mechanisms such as anti-inflammatory action [33] and central antihyperalgesic effect [34] have been suggested in adult patients, but many factors remain unknown, especially in pediatric patients.

In pediatric patients, a large number of studies reported pharmacological prevention of POV/PONV, especially with ondansetron [35,36] and dexamethasone [7,8,37]. Although the anti-PONV effects of these drugs have been established, several disadvantages have also been pointed out. Ondansetron is expensive and prolongs QT interval [9]. Dexamethasone use in certain surgeries is associated with an increased risk of postoperative bleeding [8,38]. Intravenous lidocaine is considered inexpensive and safe, and has been commonly used as an adjuvant for pediatric general anesthesia [10–13]. Lidocaine has been shown to be effective in preventing opioid-induced cough, laryngospasm, and propofol-induced pain [10–12]. In the included trials, serious adverse events of lidocaine such as seizures, arrhythmia, or allergic reactions did not occur. One trial reported there was a statistically significant difference in the time to extubation (2.5 min longer in lidocaine group) [24]; however, another trial reported there was no statistically significant difference in the duration of recovery room stay, and the time for recovery of full alertness (4-point grade scales were used) [28]. No other events of oversedation were evaluated in other trials. Thus, the use of intravenous lidocaine could be considered for prevention of POV/PONV due to its low cost, safe drug profile, and the additional effects.

In our meta-analysis, the bolus dose of lidocaine did not vary (1.5 to 2 mg.kg$^{-1}$). One trial [24] used continuous lidocaine infusion, and the calculated total amount of administered lidocaine was 2.89 mg.kg$^{-1}$. This was the highest amount of lidocaine in our included trials. However, even in this trial, the measured lidocaine concentration was 0.87 to 4.88 μg.ml$^{-1}$ [24], which was below the toxicity threshold of 5μg.ml$^{-1}$ [29]. The dose range of intravenous lidocaine used in our included trials were similar to other previous trials with pediatric patients [10–12]. We could not determine the optimal dose from our results. There is a possibility that lower intravenous lidocaine doses could be just as effective in preventing PONV, and further trials evaluating the minimal effective dose would be interesting.

Our study has several limitations. First, we could include only a small number of randomized clinical trials. We did not limit our search to trials in which the incidence of POV/PONV was the primary outcome, however, we found only 6 randomized clinical trials that evaluated the incidence of POV/PONV, and 4 out of 6 of these trials are at high risk of bias. The number of patients did not reach RIS, and the possibility of publication bias cannot be denied. Thus, we downgraded the quality of evidence to "very low". Second, we could not evaluate adverse events sufficiently because we included only randomized clinical trials that compared the incidence of POV/PONV between intravenous lidocaine and control groups. To evaluate adverse events sufficiently, we need to evaluate all trials, not only randomized controlled trials but also observational studies.

In conclusion, our meta-analysis suggests that intravenous lidocaine may reduce the incidence of POV/PONV and the need for antiemetic rescue medication in pediatric patients undergoing general anesthesia. However, the quality of the evidence was very low, and further trials with low risk of bias are necessary.

## Supporting information

**S1 Table. PRISMA checklist.**
(DOCX)

**S2 Table. Summary of findings table.**
(DOCX)

**S1 Text. Search strategy for PubMed.**
(DOCX)

## Acknowledgments

We would like to thank Editage (www.editage.com) for English language editing.

## Author Contributions

**Conceptualization:** Daisuke Nakajima, Hiromasa Kawakami.

**Data curation:** Daisuke Nakajima, Hiromasa Kawakami.

**Formal analysis:** Daisuke Nakajima, Hiromasa Kawakami.

**Investigation:** Daisuke Nakajima, Hiromasa Kawakami.

**Methodology:** Daisuke Nakajima, Hiromasa Kawakami.

**Project administration:** Daisuke Nakajima.

**Resources:** Daisuke Nakajima, Hiromasa Kawakami.

**Supervision:** Hiromasa Kawakami, Takahiro Mihara, Hitoshi Sato, Takahisa Goto.

**Validation:** Hiromasa Kawakami, Takahiro Mihara, Hitoshi Sato, Takahisa Goto.

**Visualization:** Daisuke Nakajima, Hiromasa Kawakami.

**Writing – original draft:** Daisuke Nakajima, Hiromasa Kawakami.

**Writing – review & editing:** Daisuke Nakajima, Hiromasa Kawakami, Takahiro Mihara, Hitoshi Sato, Takahisa Goto.

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
