## [Decision Letter · Decision Letter 0]

3 Jan 2020

Effectiveness of intravenous lidocaine in preventing postoperative nausea and vomiting in pediatric patients: A systematic review and meta-analysis

PONE-D-19-28099

Dear Dr. Nakajima,

We are pleased to inform you that your manuscript has been judged scientifically suitable for publication and will be formally accepted for publication once it complies with all outstanding technical requirements.

With kind regards,

Dong-Xin Wang

Academic Editor

PLOS ONE

1. We noticed minor instances of text overlap with the following previous publication(s), which need to be addressed:

https://journals.lww.com/anesthesia-analgesia/Fulltext/2019/09000/Effectiveness_of_Magnesium_in_Preventing_Shivering.14.aspx?WT.mc_id=HPxADx20100319xMP

The text that needs to be addressed involves the Abstract and 'Assessment of quality of evidence' sections.

In your revision please ensure you cite all your sources (including your own works), and quote or rephrase any duplicated text outside the methods section. Further consideration is dependent on these concerns being addressed.

'The funders had no role in study design, data collection and analysis, decision to publish, or preparation of the manuscript.'

Please provide an amended Funding Statement that declares *all* the funding or sources of support received during this specific study (whether external or internal to your organization) as detailed online in our guide for authors at http://journals.plos.org/plosone/s/submit-now

Please state what role the funders took in the study.  If any authors received a salary from any of your funders, please state which authors and which funder. If the funders had no role, please state: "The funders had no role in study design, data collection and analysis, decision to publish, or preparation of the manuscript."

c. Please send your amended statements by return email; we will change the online submission form on your behalf.

Reviewers' comments:

Reviewer's Responses to Questions

**Comments to the Author**

1. Is the manuscript technically sound, and do the data support the conclusions?

Reviewer #1: Yes

Reviewer #2: Yes

2. Has the statistical analysis been performed appropriately and rigorously? 

Reviewer #1: Yes

Reviewer #2: Yes

3. Have the authors made all data underlying the findings in their manuscript fully available?

Reviewer #1: Yes

Reviewer #2: Yes

4. Is the manuscript presented in an intelligible fashion and written in standard English?

Reviewer #1: Yes

Reviewer #2: Yes

5. Review Comments to the Author

Reviewer #1: Well presented and well researched. I'm not sure there is enough available data in the literature for this meta-analysis to further our understanding on this subject and as such I leave it to the editor to make the final decision on inclusion of this paper in this publication. From a science rigorous point of view, it's well done.

Reviewer #2: 1. POV/PONV in pediatric patients is a serious problem. Up to now, ondansetron seems to be more effective in treating and preventing PONV. The use of lidocaine may be an alternative way.

2. Although several RCT investigated the effect of lidocaine on PONV, the result is still unclear.

3. This Meta analysis provided a systematic review of presenting evidences. However, due to the low quality of these clinical trials, it's still uncertain if lidocaine dose work. This conclusion is helpful for clinical practice and further studies.

6. PLOS authors have the option to publish the peer review history of their article (what does this mean?). If published, this will include your full peer review and any attached files.

Reviewer #1: Yes: Prof. Reny Segal

Reviewer #2: No